# md_harmonize: A Python Package for Atom-Level Harmonization of Public Metabolic Databases

**DOI:** 10.3390/metabo13121199

**Published:** 2023-12-17

**Authors:** Huan Jin, Hunter N. B. Moseley

**Affiliations:** 1Department of Toxicology and Cancer Biology, University of Kentucky, Lexington, KY 40536, USA; huan.jin@regeneron.com; 2Department of Molecular & Cellular Biochemistry, University of Kentucky, Lexington, KY 40536, USA; 3Markey Cancer Center, University of Kentucky, Lexington, KY 40536, USA; 4Superfund Research Center, University of Kentucky, Lexington, KY 40506, USA; 5Institute for Biomedical Informatics, University of Kentucky, Lexington, KY 40536, USA

**Keywords:** metabolite, database harmonization, maximum common substructure, Python package

## Abstract

A major challenge to integrating public metabolic resources is the use of different nomenclatures by individual databases. This paper presents md_harmonize, an open-source Python package for harmonizing compounds and metabolic reactions across various metabolic databases. The md_harmonize package utilizes a neighborhood-specific graph coloring method for generating a unique identifier for each compound via atom identifiers based on a compound’s chemical structure. The resulting harmonized compounds and reactions can be used for various downstream analyses, including the construction of atom-resolved metabolic networks and models for metabolic flux analysis. Parts of the md_harmonize package have been optimized using a variety of computational techniques to allow certain NP-complete problems handled by the software to be tractable for these specific use-cases. The software is available on GitHub and through the Python Package Index, with end-user documentation hosted on GitHub Pages.

## 1. Introduction

Metabolic reprogramming has been recognized as a phenotypic hallmark of cancer [1]. Large-scale characterization of the metabolic phenotypes of cancers can help clarify biological mechanisms of metabolic diseases and develop novel and effective therapeutics [2]. Stable isotope tracing is an important tool for deciphering metabolic mechanisms by demultiplexing metabolic fluxes [3]. With the rapid development of analytical methodologies, especially nuclear magnetic resonance spectroscopy (NMR) and mass spectrometry (MS) [4], large volumes of high-quality isotopic-specific metabolic profiles are being generated. To derive meaningful biological interpretations from these metabolic datasets, a necessary first step is to construct reliable metabolic models [5] via a comprehensive atom-resolved metabolic network, which requires the harmonization of compounds and atom-resolved reactions from various public metabolic resources [6]. Metabolic databases, like KEGG (Kyoto Encyclopedia of Genes and Genomes) [7] and MetaCyc [8,9], contain either atom transformation patterns between reactant–product pairs [10] or direct atom mappings for reactions [11], which greatly contribute to the construction of an atom-resolved metabolic network. However, the lack of a uniform identity, especially for the atom identifiers, is a big challenge in integrating publicly available metabolic databases [12,13,14,15].

Recently, a neighborhood-specific graph coloring method was developed to generate unique identifiers for every atom and the corresponding compound based on their chemical structure [12]. This method only requires the compound molfile representation, which is available in most metabolic and chemical databases. These unified compound identifiers can facilitate both compound and reaction harmonization across public metabolic databases.

The maximum common substructure (MCS) of two compounds, a well-documented problem in graph theory, is used to search the largest substructure contained in both compound structures. The maximum common substructure method is heavily used in compound harmonization for two major purposes in the neighborhood-specific graph coloring method. One purpose is to detect the aromatic substructure(s) in a compound for structure curation, and the other is to harmonize compound pairs containing undefined generic group(s). Since MCS is a NP-complete problem, determining the maximum common substructure can be very hard to achieve in a practical amount of time for complex compounds, which can greatly limit MCS application in large-scale analyses, like metabolic database harmonization.

Here, we developed the Python package md_harmonize, utilizing a neighborhood-specific graph coloring method to facilitate the harmonization of compounds and atom-resolved metabolic reactions across various metabolic databases. Very importantly, we greatly optimized the substructure detection method by decreasing the search space via incorporating atom colors generated by neighborhood-specific graph coloring and the shortest distance between any two atoms. This package integrates data standardization and aromatic substructure detection, as well as compound and reaction harmonization together. The package also provides a relatively user-friendly command-line interface for easier incorporation into complex data analyses.

## 2. Materials and Methods

### 2.1. Compound and Metabolic Reaction Data

All data were downloaded directly from the corresponding databases. The KEGG COMPOUND and KEGG REACTION data was downloaded on 1 November 2022 (https://www.genome.jp/kegg/). MetaCyc compound and reaction data are from version 23.0 of the database, downloaded from BioCyc (https://metacyc.org/). HMDB compound data are from version 5.0 of the database (https://hmdb.ca/).

### 2.2. Matrix Representation of a Compound Structure

For a compound with n non-hydrogen atoms, we can represent its structure using an adjacent matrix N of size n × n. N(i,j) represents the bond between atom i and atom j in the compound. We adopted the standard integer representation of bond type specified in the cTfile formats [16] (Table 1). The matrix representation of KEGG compound C00207 is shown in Figure 1B.

### 2.3. Mapping Matrix between Two Compound Structures

To detect whether compound A (n non-hydrogen atoms) is a substructure of compound B (m non-hydrogen atoms), the first step is to construct a mapping matrix M with size n x m, where M(I, j) indicates whether atom i in compound A can be mapped to atom j in compound B (0 = no valid mapping; 1 = valid mapping).

Figure 2C shows the mapping matrix between KEGG compound C00207 and KEGG compound C00466 under three mapping criteria: (1) atom i in C00207 and atom j in C00466 are of the same atom type; (2) whether both atom i and atom j are in the ring structure; and (3) the number of each bond type connecting to atom i is less than or equal to the number of corresponding bond types connecting to atom j.

### 2.4. Backtracking Algorithm to Generate One-to-One Atom Mapping

A backtracking algorithm is used to generate one-to-one atom mapping for two compound structures. The flowchart of the backtracking algorithm is shown in Figure 3, with an example of detecting whether compound A (n non-hydrogen atoms) is a substructure of compound B (m non-hydrogen atoms). In addition to the two matrix representations for compound N_A_ and compound N_B_, as well as the mapping matrix M_AB_, two supplementary arrays, the one-to-one mapping array O2OM, and a “used atom” array U, are needed by the backtracking algorithm. The O2OM array is of size 1 × n, where O2OM[i] stores the index of the mapped atom in compound B for atom i in compound A. The size of the U array is 1 × m, where U[i] indicates whether atom i in compound B has been used to map an atom in the compound A (0 = not used; 1 = used). For each atom i in compound A (starting from index 1), the algorithm searches in order, testing whether there is an atom j (saved in O2OM[i]) in compound B that has not been used in the mapping so far and can pair to atom i. If one possible pair is found, then the algorithm validates the new atom pair by checking whether all the previously atom pairs that are directly connected to the new atom pair are connected the same way in the two structures. The algorithm saves the one-to-one mapping array O2OM when we find a pairing atom in compound B for every atom in compound A. Since compound B can contain multiple substructures of compound A, the algorithm will keep searching after saving the current one. When the algorithm cannot find a pairing atom j in compound B for atom i in compound A (O2OM[i] > m), the algorithm goes back to the previous atom i-1 and searches for atom i-1 from index O2OM[i-1] + 1 in compound B. During the backtrack, if the index of mapped atom for atom i-1 equals m, then the algorithm keeps moving back to atom i-2.

### 2.5. Shortest Distance between Any Two Atoms in a Compound Structure

We used the Floyd–Warshall algorithm [17] to calculate the shortest distance between any two atoms in the compound structure. Figure 4B shows the shortest distance matrix D of KEGG compound C00466.

### 2.6. Shortest Distance to the R Groups in a Compound Structure

Next, we customized Dijkstra’s algorithm [18] to calculate the shortest distance from every atom to R groups in a compound. The algorithm workflow is shown in Figure 5. This modified algorithm first initializes a distance array DR with infinity values, and a priority queue with all the R atom indeces in a compound with an associated 0 distance. Each time, an atom i with the shortest distance d is popped out from the prioirty queue, and the shortest distance of atom i is updated. The algorithm then searches for the neighboring atoms of atom i. If the neighboring atom j has DR[j] longer than d + 1, then we can further decrease the shortest distance between atom j and the R groups by pushing atom j into the priority queue. Figure 6B shows the shortest distance from each atom to the R groups in KEGG compound C05205.

### 2.7. Implementation Details of the md_hamonize Python Package

The md_harmonize package is implemented in major version 3 of the Python programming language, with an object-orientated design and following a Pythonic style. Certain computationally intensive steps were Cythonized for efficient execution. To further speed up the calculations, md_harmonize incorporates a thread and process management Pebble package [19], which enables the efficient ulitization of all available CPU cores.

## 3. Results

### 3.1. md_harmonize Package Overview

As shown in Figure 7, the md_harmonize Python package is composed of several modules. The compound.py module defines the basic elements for compound construction, composed of the ‘Atom’, ‘Bond’, and ‘Compound’ classes. The aromatics.py module contains the ‘AromaticManager’ class that is responsible for aromatic substructure construction and detection. The algorithm Biochemically Aware Substructure Search (BASS) [20] for aromatic substructure detection is implemented into the ‘BASS.pyx’ module. The ‘reaction.py’ module contains the Reaction class used to represent reactions. The ‘harmonization.py’ module takes charge of harmonizing compounds as well as reactions. The harmonized results are represented by ‘HarmonizedCompoundEdge’ and ‘HarmonizedReactionEdge’ objects. The ‘__main__.py’ module provides the command-line interface to perform data standardization, aromatic substructure detection, and compound and reaction harmonization, which is implemented with the ‘docopt’ Python library. The other modules, ‘tools.py’, ‘KEGG_database_scraper.py’, ‘KEGG_parser.py’, ‘MetaCyc_parser.py’, ‘openbabel_utils.py’, and ‘supplement.py’, define auxiliary tools for reading and writing files, scraping and parsing data, standardizing molfile representation via Open Babel [21], and processing variables.

### 3.2. md_harmonize Package Interface

The md_harmonize package provides a simple command-line interface to perform data standardization, aromatic substructure detection, and compound and reaction harmonization. Figure 8 shows version 1.0 of the command-line interface.

### 3.3. Optimization of Substructure Detection

The Biochemically Aware Substructure Search (BASS) method is used for substructure detection in the neighborhood-specific graph coloring method. Two major steps are involved in the BASS method. The first step is to construct a mapping matrix between two compound structures indicating whether an atom i in one compound can be mapped to an atom j in the other compound (Section 2.3). The second step is to find the one-to-one atom mappings of the substructures based on the mapping matrix using the backtracking algorithm (Section 2.4). The time complexity of the backtracking algorithm can hinder its application in analyzing complex or large compounds. Here, we optimized the BASS method by incorporating atom colors derived via the neighborhood-specific graph coloring method and the shortest distance between any two atoms.

We stick to the example of detecting whether compound A (n non-hydrogen atoms) is a substructure of compound B (m non-hydrogen atoms). In this case, n should be less than or equal to m. Three criteria (Section 2.3) were adopted when generating the mapping matrix between two compound structures. To fully make use of the chemical characteristics of compounds, we add another criterion based on the deduction process that the structure of compound B is constructed by adding non-hydrogen atoms/bonds to the structure of compound A. We compared the atom and bond types/counts connecting to the target atom, i.e., the number of each atom/bond types connecting to atom i in compound A cannot be larger than the number of corresponding atom/bond types of atom j in compound B. This rule applies to all cases in the substructure detection.

Another difficult case is the harmonization of generic compounds with R groups, due to the undetermined compound structures. In the KEGG COMPOUND database and MetaCyc compound database, generic compounds account for about 8% and 22%, respectively [14]. If compound A and compound B are a generic compound pair, and compound A is more generic, then the structure of compound A (ignoring R groups) is contained in the structure of compound B, and each unmatched branch in compound B structure can be mapped to an R group in compound A. Therefore, for each atom i in compound A, its chemical surroundings should be the same with that of the corresponding atom j in compound B until it meets the R groups. One advantage of the neigborhood-specific graph coloring method is that it can generate atom color layer by layer. Here, we can make use of the shortest distance to R groups d_R_ for each atom (Section 2.6), and check whether atom i in compound A and atom j in compound B have the same atom color at the d_R_-1 level.

We further optimized the backtracking alogrithm based on the axiom that incorporation of new atoms/bonds in a structure cannot increase the shortest distance between any two atoms in the original compound structure. In the previous backtracking algorithm (Section 2.4), only whether adjacent atoms are connected to the new paired atoms the same way in the corresponding compound structure was considered. To decrease the search space, we added another criterion by comparing the shortest distance between atom i and all previously mapped atoms k in the compound A to the corresponding distance in the compound B (Fomula 1). This criterion also applies to all cases in the substructure detection.
D_A_[i][k] ≥ D_B_[O2OM[i]][O2OM[k]], 1 ≤ k ≤ i-1 (1)

Here, we evaluated the algorithm performance by detecting the aromatic substructures. We subsampled 1000 compounds from the KEGG compound database and tested against the extracted 387 aromatic substructures from KEGG KCF files [12]. Next, we ran an aromatic substructure search on these 1000 compounds with and without our optimizations on a computer with an Intel(R) Core(TM) i7-6850K CPU @ 3.60 GHz with 64 GB of RAM and running the Fedora 36 Linux operating system. With optimization, the time used for the aromatic substructure search was shortened from 12,737 s to 52 s, improving by over 244 times, i.e., over a 24,400% improvement in the execution speed of the aromatic substructure search (Figure 9). However, we would like to point out that algorthim performance heavily depends on the specific compound structures being analyzed, so the level of improvement will vary depending on the set of structures analyzed. In particular, the detection of aromatic substructures in large compounds with complex fused ring structures requires extensive calculations even with our optimizations, suggesting that the algorithm can make further improvements in these difficult cases.

### 3.4. Application of md_harmonize to Compound Harmonization across Public Databases

In our prior work, we demonstrated compound and reaction harmonization of KEGG and MetaCyc using a predecessor prototype of md_harmonize [14]. Here, we used three major metabolic databases, KEGG, MetaCyc, and Human Metabolome Database (HMDB), to evaluate the results of compound harmonization generated via md_harmonize. Some compounds in HMDB have direct KEGG and/or MetaCyc references. We extracted all the HMDB-KEGG and HMDB-MetaCyc compound pairs from HMDB, then tested if those pairs can be detected by md_harmonize. The results are shown in Table 2. Based on the direct references of HMDB, 6814 KEGG-HMDB compound pairs and 2652 MetaCyc-HMDB compound pairs were detected. With md_harmonize, we discovered 8644 KEGG-HMDB compound pairs and 7271 MetaCyc-HMDB compound pairs. About 5358 KEGG-HMDB compound pairs and 1868 MetaCyc-HMDB compound pairs can be cross-validated, indicating that md_harmonize was able to detect thousands more compound pairs.

On the other hand, the md_harmonize method missed about 1456 KEGG-HMBD compound pairs and 784 MetaCyc-HMDB compound pairs indicated by HMDB references. We further investigated the causes of missed detection (Table 3). About 232 KEGG compound references and 111 MetaCyc compound references are invalid, with either no molfile representation or an incorrect compound identifier provided. We then compared the compound molecular formulas and found that 793 KEGG-HMDB compound pairs and 557 MetaCyc-HMDB compound pairs have inconsistent molecular formulas, clearly indicating that these database cross-references in HMDB are incorrect.

The remaining 431 KEGG-HMDB compound pairs and 116 MetaCyc-HMDB compound pairs are likely caused by inconsistent compound structures, as illustrated in Figure 10. Also, circular–linear interchangeable structures would not be detected, since md_harmonize requires reaction descriptions from both databases for reliably detecting such pairs, and HMDB does not have the required reaction descriptions.

## 4. Discussion

The harmonization of public metabolic databases still remains a major challenge due to the different nomenclatures used by different databases. The use of chemical identifiers like the Union of Pure and Applied Chemistry (IUPAC) International Chemical Identifier (InChI) [22,23,24] helps, but InChI does not handle ambiguously defined compound entries representing a family of chemical structures when they include R groups and repeating units. Here, we developed the md_harmonize Python package based on the neighborhood-specific graph coloring method, which provides data structures and algorithms for integrating the compound and reaction entries in public metabolic databases at atom resolution. The md_harmonize package handles both specific and ambiguously defined compound and reaction entries, with the ability to even harmonize linear and circular forms of compounds when both compound and reaction entries are present. The methods utilized by the package can also identify incorrect atom mappings across reactions and inconsistent mappings between harmonized reactions [14]. We further optimized the substructure detection method by incorporating chemical features from the compound to reduce the search space, making the NP-complete problem tractable in this use-case, as demonstrated by the 24,400% improvement in computational efficiency. As demonstrated by the added HMDB functionality of md_harmonize, the modular design, common data structures, and database parsing functionality enable highly flexible extensions to be added to support additional metabolic and compound databases. Additionally, the md_harmonize package is useful for identifying erroneous cross-database references as highlighted by Table 3 and Figure 10. Moreover, the database harmonization results generated via md_harmonize can be easily used to construct integrated metabolic networks and associated atom-resolved metabolic models. Such networks and models will better represent a larger part of the “known” metabolism, since the major metabolic network databases like KEGG and MetaCyc only partially overlap [12,14,25]. The md_harmonize Python package is available on GitHub and through the Python Package Index (PyPI), with end-user documentation on GitHub Pages (https://github.io/).

## Figures and Tables

**Figure 1 metabolites-13-01199-f001:**
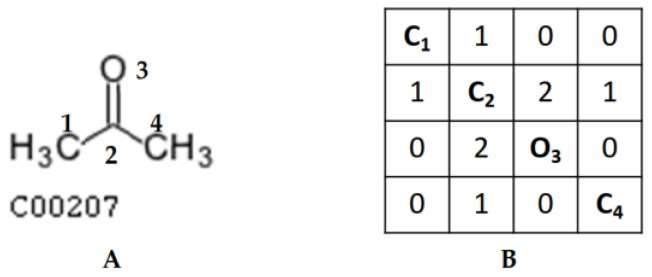
Example of matrix representation of a compound structure: (**A**) KEGG compound C00207 with the atoms numbered for comparison to rows and columns in the matrix; (**B**) matrix representation of KEGG compound C00207.

**Figure 2 metabolites-13-01199-f002:**
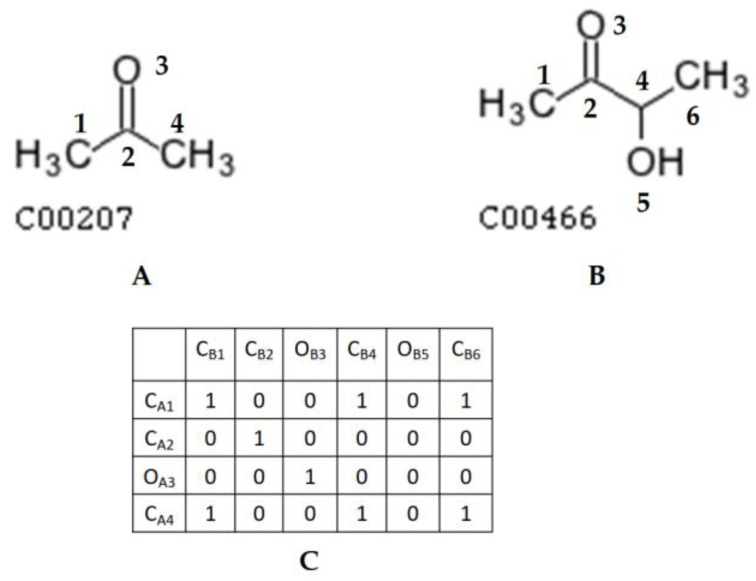
Example of a mapping matrix between two compound structures: (**A**) KEGG compound C00207 with atoms numbered to rows in the matrix; (**B**) KEGG compound C00466 with atoms numbered to columns in the matrix; and (**C**) mapping matrix between KEGG compound C00207 and KEGG compound C00466.

**Figure 3 metabolites-13-01199-f003:**
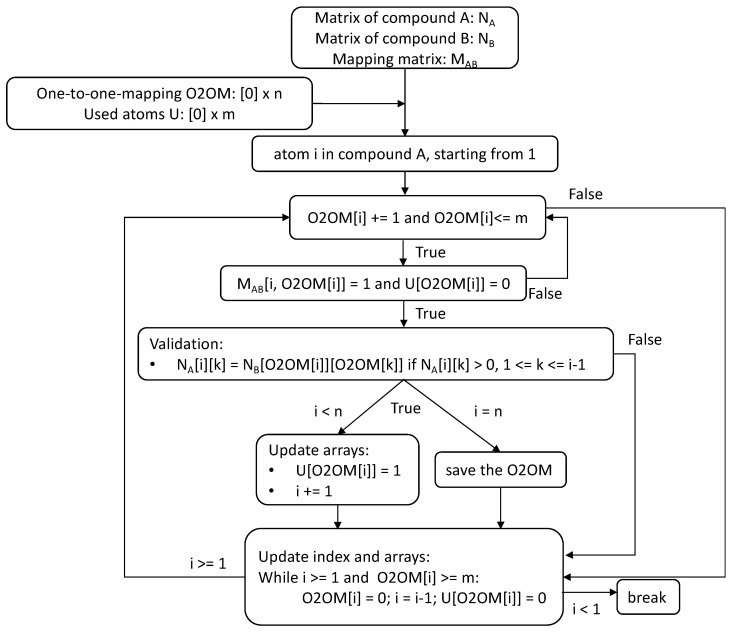
Flowchart of backtracking algorithm for generating one-to-one atom mappings of two compound structures.

**Figure 4 metabolites-13-01199-f004:**
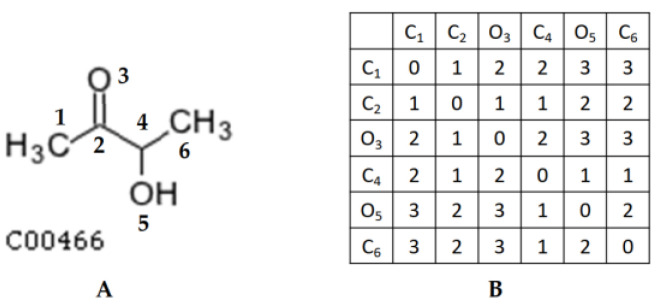
Example of the shortest distance between any two atoms in a compound structure: (**A**) KEGG compound C00466 with atoms numbered to rows and columns in the matrix; (**B**) the shortest distance matrix D of KEGG compound C00466.

**Figure 5 metabolites-13-01199-f005:**
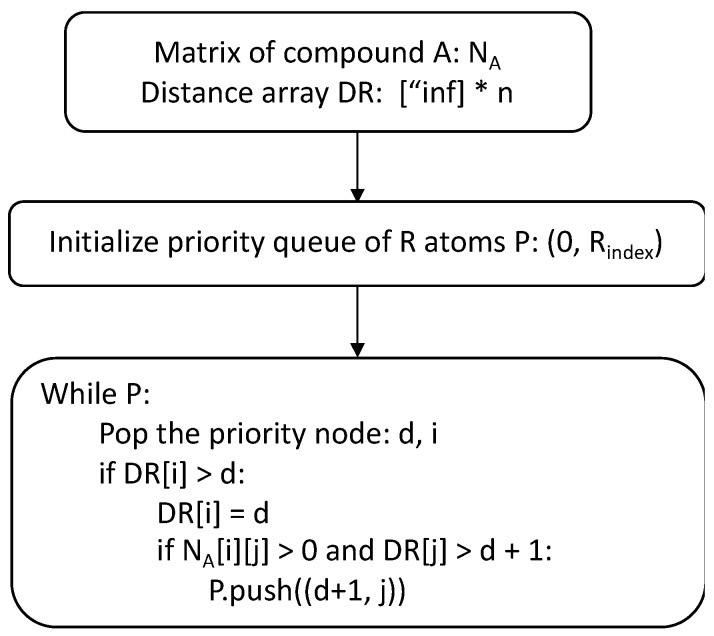
Flowchart of the modified Dijkstra algorithm for generating the shortest distance between each atom and the R groups in a compound. The “*” represents the multiplication operator.

**Figure 6 metabolites-13-01199-f006:**
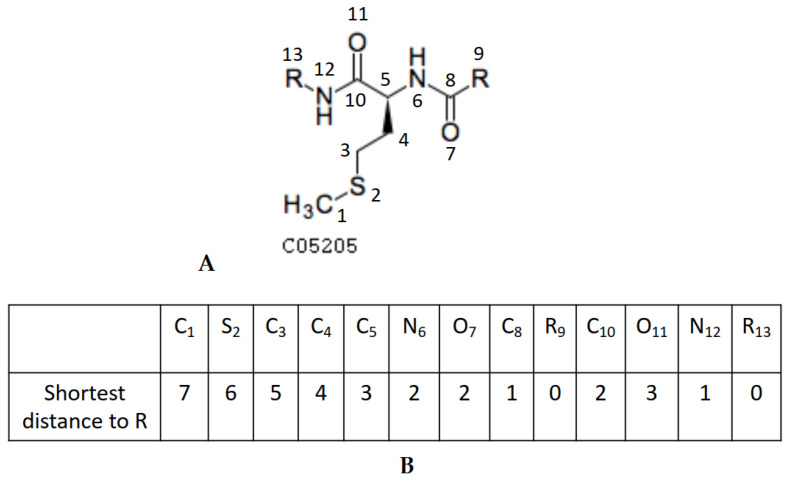
Shortest distance to the R groups in a compound structure. (**A**) KEGG compound C05205 with atoms numbered to indeces in the array; (**B**) the array of the shortest distance from each atom to R groups in KEGG compound C05205.

**Figure 7 metabolites-13-01199-f007:**
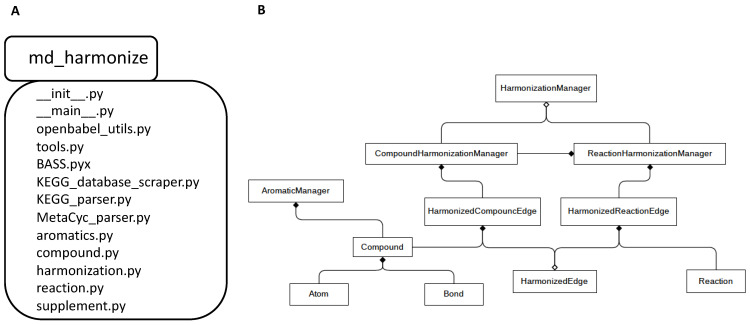
Organization of the md_harmonize package presented with UML diagrams. (**A**) UML package diagram of the md_harmonize Python library; (**B**) UML class diagram of the md_harmonize Python package.

**Figure 8 metabolites-13-01199-f008:**
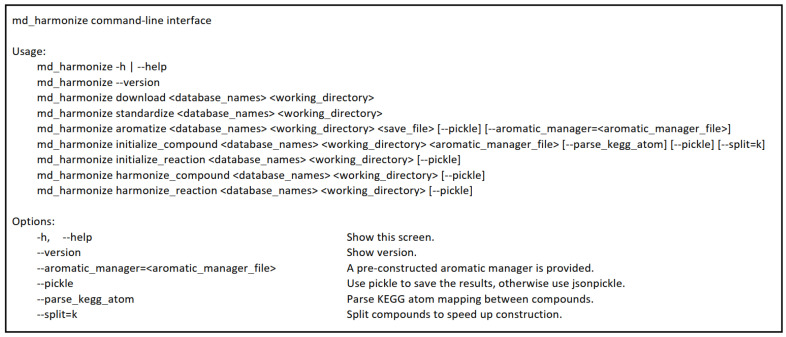
Command line interface of md_harmonize package.

**Figure 9 metabolites-13-01199-f009:**
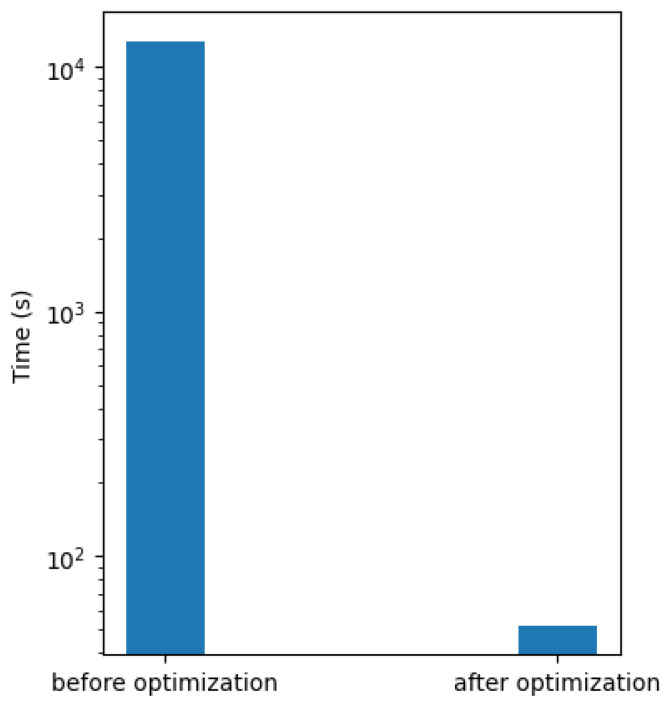
Comparison of substructure performance after algorithm optimization.

**Figure 10 metabolites-13-01199-f010:**
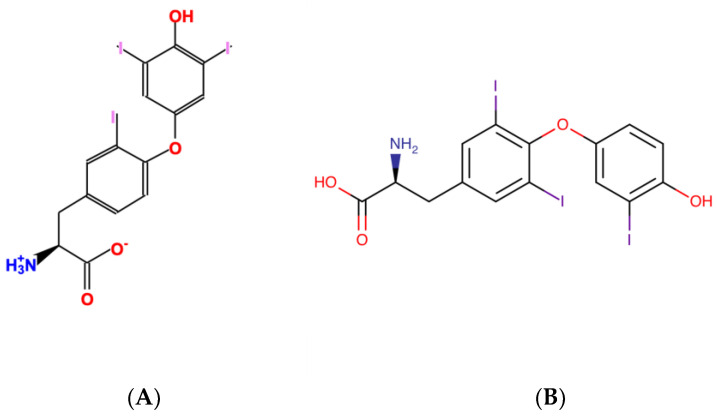
Example of incorrect compound pair indicated via HMDB reference with different structure representations. (**A**) MetaCyc CPD-10813; (**B**) HMDB HMDB0000265.

**Table 1 metabolites-13-01199-t001:** Integer representation of bond type.

Bond Type	Integer
1	Single
2	Double
3	Triple
4	Aromatic

**Table 2 metabolites-13-01199-t002:** Comparison of compound harmonization.

Databases	HMDB Extracted	md_harmonized Detected	Overlap
KEGG	6814	8644	5358
MetaCyc	2652	7271	1868

**Table 3 metabolites-13-01199-t003:** Categorization of compound pairs missed by md_harmonize.

Category	KEGG	MetaCyc
Invalid references	232 (15.93%)	111 (14.15%)
Inconsistent formulas	793 (54.46%)	557 (71.05%)
Other	431 (29.61%)	116 (14.80%)
Total	1456	784

## Data Availability

All generated results in this manuscript are available at: https://doi.org/10.6084/m9.figshare.21699683. The documentation for the md_harmonize Python package is available at: https://moseleybioinformaticslab.github.io/md_harmonize/.

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
