# Peer review of "md_harmonize: A Python Package for Atom-Level Harmonization of Public Metabolic Databases"

_metabolites, 2023, doi:10.3390/metabo13121199_

Round 1
Reviewer 1 Report
Comments and Suggestions for Authors
Please avoid bulk citation (line [9-12]). Use more recent access date to major databases. Check figure quality for chemical formulas.
Line 36: “KEGG (Kyoto Encyclopedia of Genes and Genomes) and MetaCyc” – add references for KEGG and MetaCyc
Lines 68-69 - add the references (online links) to these databases, correspondingly
Use Italic font for formulas and variables ‘n’m, ‘I’, ‘j’ - (“size n x n” , “atom I”) in the text and I the formulas ( like 1)
Line 293: “24400% improvement” – please rephrase
Author Response
Reviewer 1:
Please avoid bulk citation (line [9-12]). Use more recent access date to major databases. Check figure quality for chemical formulas.
Response:
The references [9-12] at line 40 all point out the challenge with integrating publicly-available metabolic databases. But we felt it was important to highlight that several prior publications had identified this challenge, which our md_harmonize software package addresses. This is the only place we used a bulk citation.
With respect to access dates, November 2022 is when we accessed KEGG and pulled down the data used to generate our results. For MetaCyc and HMDB, we indicated the version of the database used.
We assume the reviewer meant chemical structures in the figures and not chemical formulas, since we have no chemical formulas in the figures. We see no major problem with the chemical structures, except maybe Figure 10. We have increased the size of the chemical structures in Figure 10.
Issue 1:
Line 36: “KEGG (Kyoto Encyclopedia of Genes and Genomes) and MetaCyc” – add references for KEGG and MetaCyc
Response:
Thanks for catching this! We have added these references along with a reference for the HMDB.
Issue 2:
Lines 68-69 - add the references (online links) to these databases, correspondingly
Response:
Again, thanks for catching this! We have added these references.
Issue 3:
Use Italic font for formulas and variables ‘n’m, ‘I’, ‘j’ - (“size n x n” , “atom I”) in the text and I the formulas ( like 1)
Response:
Thanks for catching this at line 75! We have double-checked and updated the formulas and variables.
Issue 4:
Line 293: “24400% improvement” – please rephrase
Response:
We have rephrased as follows:
“With optimization, the time used for aromatics substructure search was shortened from 12737s to 52s, improving by over 244 times, i.e. over 24,400% improvement in the execution speed of the aromatic substructure search (Figure 9).”
Reviewer 2 Report
Comments and Suggestions for Authors
The authors present a Python package for harmonization of public metabolic databases. The manuscript is very interesting, but it is unclear what the 'average' reader is supposed to do with the information. As it stands, the manuscript lacks proper worked examples or links to documentation or tutorials, which would help the 'average' reader to understand the importance of the work.
GENERAL COMMENTS
If my understanding of the manuscript is correct, the proposed Python package provides alternative mappings of HMDB and KEGG and MetaCYC identifiers based on atom and bond mapping. It might be helpful to explicitly tabulate the number of false positives in existing mappings in summary format, and the number of false negatives (almost like a confusion matrix?). It also seems relevant to provide in Supplementary Information an explicit list of the new mappings, and the false positive mappings. For the typical user, this seems more helpful than simply providing the code so that users can rerun the package themselves?
Presumably the paper is also targeted at the administrators of the databases. What actions do the authors recommend that the databases take, if indeed the current mappings are incorrect?
It would also be very helpful to have some examples in each category. For example, pick a compound that is not currently mapped between two databases, and show how md_harmonize maps correctly. Similarly, show some mappings that are currently incorrect, and demonstrate that md_harmonize identifies these 'false positive' mappings.
Finally, as a non-expert in this area, I am surprised no-one has attempted such a mapping before. What methods and approaches do the database owners use for their mappings? Is there scope for md_harmonize to be adopted by the databases? It would be helpful if the Introduction could do more to explain why the proposed package is novel, what unmet need it fulfills, and why existing approaches perform so (apparently) poorly.
Why no tutorials, or even links to tutorials? I was able to google this:
Welcome to MDH’s documentation! — md_harmonize 1.0.4 documentation (moseleybioinformaticslab.github.io)
Ideally, supporting information on a Python library should be available when reading a paper on said library.
Finally, the Discussion section is far too short. The authors make a lot of claims that md_harmonize will help make metabolomics easier, but as a non-expert user, I am confused as to how this is the case. There are no Results demonstrating these conclusions (see also specific comment on Line 297 below).
SPECIFIC COMMENTS
Line 74, to my knowledge 'heavy atoms' can be used to describe atoms above C and N, and so is somewhat ambiguous here. It might be better for the authors to say 'non-hydrogen atoms' instead
Line 227, mis-spelled 'further'
Figure 9 is superfluous, I think. What additional information do the authors think that Figure 9 conveys? Does the Figure replace a disproportionate number of words, or make a difficult concept easier to understand, or represent a complex dataset in an efficient way?
Line 297, 'Additionally, the database harmonization results generated by md_harmonize can be easily used to construct integrated metabolic networks and associated atom-resolved metabolic models'
Can it be easily used in this way? How? Are there any instructions on how to do this? Can the authors give an example of constructing an integrated metabolic network and show that md_harmonize assists in this process? Or is it better to make a more general statement, that harmonization of the databases and their mappings will help comparison of metabolomics work by providing more commonality.
Comments on the Quality of English LanguageThe standard of written language is very good and few errors can be found.
Author Response
Reviewer 2:
The authors present a Python package for harmonization of public metabolic databases. The manuscript is very interesting, but it is unclear what the 'average' reader is supposed to do with the information. As it stands, the manuscript lacks proper worked examples or links to documentation or tutorials, which would help the 'average' reader to understand the importance of the work.
Response:
We thank the reviewer for their positive comments. Also, we thank the reviewer for pointing out the need for links to the end-user documentation. We have added these links to the documentation into the Data Availability Statement. We have also highlighted this both in the Abstract and the Discussion section:
“The md_harmonize Python package is available on GitHub and through the Python Package Index (PyPI), with end-user documentation on GitHub Pages (github.io).”
Also, the Figshare item includes a simple example of using the package.
Issue 1:
GENERAL COMMENTS
If my understanding of the manuscript is correct, the proposed Python package provides alternative mappings of HMDB and KEGG and MetaCYC identifiers based on atom and bond mapping. It might be helpful to explicitly tabulate the number of false positives in existing mappings in summary format, and the number of false negatives (almost like a confusion matrix?). It also seems relevant to provide in Supplementary Information an explicit list of the new mappings, and the false positive mappings. For the typical user, this seems more helpful than simply providing the code so that users can rerun the package themselves?
Response:
The package creates atom-specific identifiers based on neighborhood-specific chemical graph coloring. In this manuscript, we are NOT focused on evaluating the atom mappings across reactions that are provided by KEGG and MetaCyc. What we provide is a harmonization of compounds entries across KEGG, MetaCyc, and HMDB databases. We have also included an evaluation of cross-database references provided by HMDB. This is one of the significant results presented in the manuscript. All results are present in the supplemental material, which is a Figshare item. We did not include a harmonization of reactions, because HMDB does not provide atom-resolved reactions entries.
Issue 2:
Presumably the paper is also targeted at the administrators of the databases. What actions do the authors recommend that the databases take, if indeed the current mappings are incorrect?
Response:
In prior publications where we have shown issues in databases and repositories, the database/repository typically has addressed most or at least some of the issues. Please see our recent publications demonstrating issues in databases/repositories:
Erik Huckvale and Hunter N.B. Moseley. "kegg_pull: a Software Package for the RESTful Access and Pulling from The Kyoto Encyclopedia of Gene and Genomes" BMC Bioinformatics 24, 78 (2023).
Christian D. Powell and Hunter N.B. Moseley. "The Metabolomics Workbench File Status Website: A Metadata Repository Promoting FAIR Principles of Metabolomics Data" BMC Bioinformatics 24, 299 (2023).
Christian D. Powell and Hunter N.B. Moseley. "The mwtab Python library for RESTful Access and Enhanced Quality Control, Deposition, and Curation of the Metabolomics Workbench Data Repository" Metabolites 11, 163 (2021).
Huan Jin and Hunter N.B. Moseley. "Hierarchical Harmonization of Atom-Resolved Metabolic Reactions Across Metabolic Databases" Metabolites 11, 431 (2021).
Issue 3:
It would also be very helpful to have some examples in each category. For example, pick a compound that is not currently mapped between two databases, and show how md_harmonize maps correctly. Similarly, show some mappings that are currently incorrect, and demonstrate that md_harmonize identifies these 'false positive' mappings.
Response:
We have already done this in prior publications:
Huan Jin and Hunter N.B. Moseley. "Hierarchical Harmonization of Atom-Resolved Metabolic Reactions Across Metabolic Databases" Metabolites 11, 431 (2021).
Huan Jin, Joshua M. Mitchell, and Hunter N.B. Moseley. "Atom Identifiers Generated by a Neighborhood-Specific Graph Coloring Method Enable Compound Harmonization Across Metabolic Databases" Metabolites 10, 368 (2020).
We have added a sentence pointing out examples of incorrect mappings detected in prior methods publications:
“The methods utilized by the package can also identify incorrect atom mappings across reactions and inconsistent mappings between harmonized reactions [14].”
On a related issue, Table 3 and Figure 10 highlight incorrect cross-database references within the HMDB, which we have further highlighted in the Discussion section now:
“Additionally, the md_harmonize package is useful for identifying erroneous cross-database references as highlighted by Table 3 and Figure 10.”
Issue 4:
Finally, as a non-expert in this area, I am surprised no-one has attempted such a mapping before. What methods and approaches do the database owners use for their mappings? Is there scope for md_harmonize to be adopted by the databases? It would be helpful if the Introduction could do more to explain why the proposed package is novel, what unmet need it fulfills, and why existing approaches perform so (apparently) poorly.
Response:
There have been multiple methods developed to create cross-reaction atom mappings. Both KEGG and MetaCyc have internal methods they have developed for creating such mappings. However, the cross-database references which this paper focuses on are another matter. Most databases have relied on InChI identifiers and the like. However, many compounds have R groups and thus have no InChI identifiers. We have highlighted this point in the Discussion section now:
“Use of chemical identifiers like the Union of Pure and Applied Chemistry (IUPAC) International Chemical Identifier (InChI) [22]-[24] helps, but InChI does not handle ambiguously defined compound entries representing a family of chemical structures when they include R groups and repeating units. Here, we developed the md_harmonize Python package based on the neighborhood-specific graph coloring method, which provides data structures and algorithms for integrating the compound and reaction entries in public metabolic databases at atom resolution. The md_harmonize package handles both specific and ambiguously defined compound and reaction entries, with the ability to even harmonize linear and circular forms of compounds when both compound and reaction entries are present.”
Besides our methods, there are no methods we know of for harmonizing both compounds and reactions across databases to this level of rigor. There is a recent paper that generates and harmonizes cross-reaction atom mappings across BRENDA, KEGG, and MetaCyc.
Starke C, Wegner A. MetAMDB: Metabolic Atom Mapping Database. Metabolites. 2022 Jan 27;12(2):122.
This paper actually utilizes our harmonization of KEGG and MetaCyc reactions.
Issue 5:
Why no tutorials, or even links to tutorials? I was able to google this:
Welcome to MDH’s documentation! — md_harmonize 1.0.4 documentation (moseleybioinformaticslab.github.io)
Ideally, supporting information on a Python library should be available when reading a paper on said library.
Response:
Thank you for pointing this out. We have fixed this shortsight. The md_harmonize documentation is now mentioned in the Abstract, the Discussion section, and the Data Availability sections:
“The md_harmonize Python package is available on GitHub and through the Python Package Index (PyPI), with end-user documentation on GitHub Pages (github.io).”
Issue 6:
Finally, the Discussion section is far too short. The authors make a lot of claims that md_harmonize will help make metabolomics easier, but as a non-expert user, I am confused as to how this is the case. There are no Results demonstrating these conclusions (see also specific comment on Line 297 below).
Response:
We agree and have significantly expanded the Discussion section to better describe the benefits provided by the md_harmonize software package.
Issue 7:
SPECIFIC COMMENTS
Line 74, to my knowledge 'heavy atoms' can be used to describe atoms above C and N, and so is somewhat ambiguous here. It might be better for the authors to say 'non-hydrogen atoms' instead
Response:
Thanks! Fixed.
Issue 8:
Line 227, mis-spelled 'further'
Response:
Thanks! Fixed.
Issue 9:
Figure 9 is superfluous, I think. What additional information do the authors think that Figure 9 conveys? Does the Figure replace a disproportionate number of words, or make a difficult concept easier to understand, or represent a complex dataset in an efficient way?
Response:
Respectfully, we disagree. This graph visually illustrates the improvement in computation performance. Some readers are visual learners and really like a graph, even a simple bar graph like Figure 9.
Issue 10:
Line 297, 'Additionally, the database harmonization results generated by md_harmonize can be easily used to construct integrated metabolic networks and associated atom-resolved metabolic models'
Can it be easily used in this way? How? Are there any instructions on how to do this? Can the authors give an example of constructing an integrated metabolic network and show that md_harmonize assists in this process? Or is it better to make a more general statement, that harmonization of the databases and their mappings will help comparison of metabolomics work by providing more commonality.
Response:
The word “easy” is relative. But without our cross-database harmonization, it is practically impossible to create metabolic models based on more than one metabolic database. We have rephrased as follows:
“Moreover, the database harmonization results generated by md_harmonize can be straight-forwardly used to construct integrated metabolic networks and associated atom-resolved metabolic models. Such networks and models will better represent a larger part of “known” metabolism, since the major metabolic network databases like KEGG and MetaCyc only partially overlap [12],[14],[25].”
Reviewer 3 Report
Comments and Suggestions for Authors
Comments:
The article presented a pioneer study of a research, the authors aimed to provide a relatively user-friendly command-line interface for easier incorporation into database analyses. The field of action of the work presents scientific relevance; especially in Metabolic reprogramming highly associated metabolic models. The authors concluded that a metabolomics approach was used to identify metabolism disorders in public health. I am interested in this topic; however, I have several comments:
1. The authors should clarify the feature and novel findings of this study.
2. Previous studies demonstrated that metabolism disorders and public health were associated to atom-level harmonization. Is there other studies which found similar result to yours study?
3. The authors should add the comments related to selection bias in this study to the perceived limitation subsection.
4. Please revise the title of your manuscript so that it contains details of the study design which characterize the investigation as well.
5. I am not familiar with package development. Please give rationale for this approach.
Author Response
Reviewer 3:
Comments:
The article presented a pioneer study of a research, the authors aimed to provide a relatively user-friendly command-line interface for easier incorporation into database analyses. The field of action of the work presents scientific relevance; especially in Metabolic reprogramming highly associated metabolic models. The authors concluded that a metabolomics approach was used to identify metabolism disorders in public health. I am interested in this topic; however, I have several comments:
Response:
We thank the reviewer for their positive comments. However, we did NOT conclude that a metabolic approach was used to identify metabolism disorders in public health. We only mentioned some prior published research where they used metabolomics and metabolic modeling to study the metabolic underpinnings of specific human diseases like cancer. This was to provide context for our software development. Our conclusions were as follows:
“As demonstrated by the added HMDB functionality of md_harmonize, the modular design, common data structures, and database parsing functionality enables highly flexible extensions to be added to support additional metabolic and compound databases. Additionally, the md_harmonize package is useful for identifying erroneous cross-database references as highlighted by Table 3 and Figure 10. Moreover, the database harmonization results generated by md_harmonize can be straight-forwardly used to construct integrated metabolic networks and associated atom-resolved metabolic models. Such networks and models will better represent a larger part of “known” metabolism, since the major metabolic network databases like KEGG and MetaCyc only partially overlap [12],[14],[25]. The md_harmonize Python package is available on GitHub and through the Python Package Index (PyPI), with end-user documentation on GitHub Pages (github.io).”
Issue 1:
- The authors should clarify the feature and novel findings of this study.
Response:
We have clarified the novel findings as follows:
“Here, we developed the md_harmonize Python package based on the neighborhood-specific graph coloring method, which provides data structures and algorithms for integrating the compound and reaction entries in public metabolic databases at atom resolution. The md_harmonize package handles both specific and ambiguously de-fined compound and reaction entries, with the ability to even harmonize linear and circular forms of compounds when both compound and reaction entries are present. The methods utilized by the package can also identify incorrect atom mappings across re-actions and inconsistent mappings between harmonized reactions [14]. We further optimized the substructure detection method by incorporating chemical features from the compound to reduce the search space, making the NP-complete problem tractable in this use-case as demonstrated by the 24,400% improvement in computational efficiency. As demonstrated by the added HMDB functionality of md_harmonize, the modular design, common data structures, and database parsing functionality enables highly flexible extensions to be added to support additional metabolic and compound databases. Additionally, the md_harmonize package is useful for identifying erroneous cross-database references as highlighted by Table 3 and Figure 10. Moreover, the database harmonization results generated by md_harmonize can be straight-forwardly used to construct integrated metabolic networks and associated atom-resolved metabolic models. Such networks and models will better represent a larger part of “known” metabolism, since the major metabolic network databases like KEGG and MetaCyc only partially overlap [12],[14],[25]. The md_harmonize Python package is available on GitHub and through the Python Package Index (PyPI), with end-user documentation on GitHub Pages (github.io).”
Issue 2:
- Previous studies demonstrated that metabolism disorders and public health were associated to atom-level harmonization. Is there other studies which found similar result to yours study?
Response:
The reviewer has missed the point of this manuscript. We present software that harmonizes metabolic databases, both compound and reaction entries. Such software enables utilization of multiple metabolic databases for a variety of purposes, including metabolic model creation for analyzing and interpreting metabolomics datasets. Many prior publications have used metabolic models derived from a single database for these purposes which only represent a part of “known metabolism”. We have tried to make these points clearer as follows:
“Moreover, the database harmonization results generated by md_harmonize can be straight-forwardly used to construct integrated metabolic networks and associated atom-resolved metabolic models. Such networks and models will better represent a larger part of “known” metabolism, since the major metabolic network databases like KEGG and MetaCyc only partially overlap [12],[14], [25].”
Issue 3:
- The authors should add the comments related to selection bias in this study to the perceived limitation subsection.
Response:
There is NO selection bias. We do not understand what the reviewer is talking about.
Issue 4:
- Please revise the title of your manuscript so that it contains details of the study design which characterize the investigation as well.
Response:
Our title is: “md_harmonize: a Python package for atom-level harmonization of public metabolic databases”. This well describes the main topic of this manuscript, which is a new Python package called md_harmonize, which provides atom-level harmonization of metabolic databases, which includes both compound and reaction entry harmonization.
Issue 5:
- I am not familiar with package development. Please give rationale for this approach.
Response:
With all due respect, if you are not familiar with computer programming, software package development, or metabolic modeling, then this manuscript is not targeted for your expertise. Sophisticated software as well as “known (cellular) metabolism” represented in complex metabolic networks within metabolic databases like KEGG and MetaCyc is required for metabolic modeling.
Round 2
Reviewer 2 Report
Comments and Suggestions for Authors
The authors have responded appropriately to the comments and I thank them for their efforts. Whilst I am unconvinced that Figure 9 is a good use of the page space, this is a minor personal preference point. I also think it would be worthwhile to have more discussion of recommendations for database providers and future work in the Discussion section, as the current state of affairs seems suboptimal (the authors' interventions are helpful but surely a better co-ordination effort between databases would be a preferable solution), but these are small suggestions and I would be content to see the article published as-is.